# How Possible Is the Elimination of Viral Hepatitis? An Analysis Based on the Global Burden of Disease from Hepatitis B and C, 1990–2019

**DOI:** 10.3390/microorganisms12020388

**Published:** 2024-02-15

**Authors:** Nelson Alvis-Guzman, Nelson J. Alvis-Zakzuk, Fernando De la Hoz Restrepo

**Affiliations:** 1Department of Economic Sciences, Universidad de Cartagena, Cartagena 130001, Colombia; 2Research Group in Hospital Management and Health Policies, Universidad de la Costa, Barranquilla 080001, Colombia; nalvis1@cuc.edu.co; 3Programa de Posgraduação em Epidemiología, Faculdade de Saúde Pública, Universidade de São Paulo, Sao Paulo 01246-904, Brazil; 4Universidad Nacional de Colombia, Bogotá 110110, Colombia; fpdelahozr@unal.edu.co

**Keywords:** hepatitis C, elimination, Global Burden of Diseases, mortality

## Abstract

This study assesses the feasibility of hepatitis B (HBV) and C (HCV) elimination using an analysis of trends of epidemiology data (1990–2019) from the Global Burden of Disease Study. Joinpoint regression analysis was used to identify significantly changing points in the trends of Age-standardized Prevalence Rates (ASPR) and Age-standardized Mortality Rates (ASMR) and to estimate the annual percentage changes (APC) and the average annual percentage changes (AAPC) for the period. The Sociodemographic Index (SDI) was used to analyze trends between countries. The total percentage change of the ASPR (2019/1990) was −31.4% and −12.8% for HBV and HCV worldwide, respectively; the rate ratio (HBV/HCV) was 2.5. Mortality had decreased for HBV but not for HCV. The total percentage change for the ASMR (2019/1990) was −26.7% and 10.0% for HBV and HCV, respectively. While the ASMR of HBV decreased, HCV increased during this period. The percentage change in ASMR of HBV was highest in countries with high–middle SDI and lowest in countries with high SDI. For HCV, the percentage change in ASMR was highest in countries with high SDI (increase), and only in countries with low SDI did it decrease. The global HBV and HCV rates have fallen with different AAPCs associated with the SDI. Despite the advances, there is still a long way to go to achieve the 2030 elimination goals. An important challenge is related to finding a way to speed up the yearly rate at which the decline is happening.

## 1. Introduction

At the beginning of the last century, the responses to outbreaks and epidemics of infectious diseases were linked to the mythical, magical, and religious conception of them [1]. By the end of the 19th century and the beginning of the 20th, the discoveries of microorganisms that cause specific diseases (tuberculosis, cholera, diphtheria, tetanus, pneumonia, meningitis, syphilis, and others) and the identification of vectors capable of transmitting diseases (the Anopheles mosquito for malaria) [2] consolidated a new paradigm for the study of infectious diseases, their control, and their prevention [3]. In addition, with the development of vaccines and the emergence of antimicrobial therapy during the first half of the 20th century, we were excited about the disappearance of infectious diseases [4]. At least, that was the perception that high-income countries had during the 1960s and 1970s, especially inspired by the eradication of smallpox [5] and which, in some way, motivated the theory of epidemiological transition [6].

The experience of the eradication of smallpox related to the mobilization of resources in public health, the learning curve in decision making, and the international alignment around common health objectives launched a new stage of discussion around setting goals for the eradication or elimination of infectious diseases. This new stage was mainly inspired by the goal of “Health for all by the year 2000” from the WHO [7]. Thus, in 1993, the Centers for Disease Control and Prevention (CDC) published a list describing the potentially eradicable diseases, based on the recommendations resulting from the International Task Force for the Eradication of Diseases (ITFDE) [8]. The aim of the ITFDE was to establish criteria to systematically assess the potential for the eradication of other diseases after the Smallpox Eradication Program. The ITFDE defined eradication as “the reduction to zero of the incidence of a disease worldwide, as a result of deliberate efforts, obviating the need for additional control measures” [8].

Based on the existence of highly effective control measures (vaccine against HBV and treatment against HCV), in 2016, WHO outlined ambitious targets to achieve its global elimination by 2030, defined as a 90% reduction in new infections and a 65% reduction in mortality [9]. “However, it is a silent epidemic because infected people do not present symptoms until the liver has already been damaged. That is why it is important for the countries to step up efforts to reach the goal of eliminating hepatitis as a public health problem in the Region by 2030”, said Massimo Ghidinelli, chief of the HIV, Hepatitis, Tuberculosis, and Sexually Transmitted Infections unit at PAHO/WHO [10].

Although the definitions of eradication and elimination of an infectious disease are clear, the issues affecting the achievement of these goals include the estimation of the existing disease burden, its current dynamics, and probable future scenarios given the current effectiveness of the respective intervention controls. The Global Burden of Diseases, Injuries, and Risk Factors Study (GBD) gathers disability and mortality data on HBV and HCV infections, from 1990 to 2019 [11]. The GBD is the single largest and most detailed scientific effort ever conducted to quantify levels and trends in health. The GBD produces regular estimates of all-cause mortality, deaths by cause, years of life lost due to premature mortality (YLLs), years lived with disability (YLDs), and disability-adjusted life years (DALYs) [12].

In the present study, we assessed the trends of the global burden of viral hepatitis (B and C) disease for sexes, ages, and countries grouped by Sociodemographic Index (SDI), to evaluate the feasibility of achieving elimination targets by 2030.

## 2. Materials and Methods

### 2.1. Study Design

We conducted an ecological study to describe trends in prevalence, incidence, and mortality from HBV and HCV infections worldwide from 1990 to 2019. We used the Sociodemographic Index (SDI) categories (high, middle–high, middle, middle–low, and low) that were defined for 195 countries and territories by calculating the geometric mean of three components: (a) total fertility rate under the age of 25 (TFU25); (b) mean education for those aged 15 and older (EDU15+); and (c) lag distributed income (LDI) per capita [11]. The SDI is a composite indicator of development status that is strongly correlated with health outcomes. It is used in the Global Burden of Disease (GBD) study to help produce estimates of the burden of diseases, injuries, and risk factors for various locations [11].

### 2.2. Data

Data were taken from the GBD 2019, a publicly available online resource “https://vizhub.healthdata.org/gbd-results/” (accessed on 17 August 2023) by the Institute for Health Metrics and Evaluation (IHME) at the University of Washington [11]. The IHME estimated prevalence, incidence, and death (number, percent, and rate) by cause, location, age, sex, year, and annual rate of change, from 1990 to 2019. The rates included were Age-standardized Prevalence Rates (ASPR), Age-standardized Incidence Rates (ASIR), and Age-standardized Mortality Rates (ASMR) for the analyzed period [11]. The GBD 2019 estimates incidence, prevalence, mortality, years of life lost (YLLs), years lived with disability (YLDs), and disability-adjusted life years (DALYs) due to 369 diseases and injuries; estimates were made for both sexes and 204 countries and territories. Input data were extracted from censuses, household surveys, civil registration and vital statistics, disease registries, health service use, air pollution monitors, satellite imaging, disease notifications, and other sources [11].

For this study, the outcomes were selected as “Total burden related to hepatitis B” and “Total burden related to hepatitis C” for both sexes and all ages.

### 2.3. Analysis

The ratio percentage changes of the ASIR, ASPR, and ASMR for HBV and HCV were estimated by dividing the corresponding value for HBV by that for HCV in order to describe the relative change.

Through Joinpoint regression analysis (segmented regression model), we identified significant changing points in trends of ASPR, ASIR, and ASMR of viral hepatitis (HBV and HCV) for the analyzed period from 1990 to 2019. This model can find and identify points where significant changes occur (breakpoints) and where the dependent variable has changed significantly at different times.

For this study, we used the “Methodology for Characterizing Trends” of the National Cancer Institute of the United States [13] that uses the Joinpoint Regression Program (Version 5.0.2. May 2023) [14]. The Joinpoint software uses statistical criteria to determine (a) the fewest number of segments necessary to characterize a trend; (b) where the segments begin and end; and (c) the annual percent change (APC) for each segment (a linear trend on a log scale implies a constant APC) and the average annual percentage change (AAPC) [13]. HBV and HCV incidence, prevalence, and mortality trends were analyzed as follows: “changing less than or equal to 0.5% per year (−0.5 ≤ APC ≤ 0.5), and the APC was not statistically significant, we characterized it as stable; changing more than 0.5% per year (APC < −0.5 or APC > 0.5), and the APC was not statistically significant, we characterized it as non-significant change; changing with a statistically significant APC > 0, we characterized it as rising; and changing with a statistically significant APC < 0, we characterized it as falling” [13].

## 3. Results

### 3.1. HBV and HCV Infection Prevalence and Incidence

In 2019, there were 439.5 million (95% [uncertainty interval] UI 384.2–500.1) people of all ages living with HBV and HCV (326.2 million [95% UI 292.2–361.1] for HBV and 113.2 million [95% UI 92.0–139.0] for HCV), worldwide. The ASPR for HBV and HCV was 4216.3 (3776.3–4667.2) and 1463.3 (1189.0–1796.6) per 100,000 people, respectively. The total percentage change of the ASPR (2019/1990) was −31.4% and −12.8% for HBV and HCV, respectively. The rate ratio (HBV/HCV) was 2.5 in the study period. The percentage change of the ASPR for HBV and HCV was highest in countries with middle SDI and lowest in countries with high SDI. For HCV, the prevalence rate increased 11.2% in high-SDI countries during the period (Table 1, Appendix A).

For 2019, the incident cases of HBV were 80.0 million (95% UI 63.9–98.2) and for HCV 5.5 million (95% UI 4.8–6.3), globally. The ASIR for HBV and HCV was 1034.2 (827.1–1270.3) and 71.3 (62.8–82.6) per 100,000 people, respectively. The total percentage change of the ASIR (2019/1990) was −34.1% and −16.8% for HBV and HCV, respectively. The rate ratio of the percentage change (HBV/HCV) was 2.04 in the study period. The percentage change of the ASIR for HBV was highest in countries with middle SDI and lowest in countries with low SDI. For HCV, it was highest in countries with middle SDI and lowest in countries with high SDI; in the latter, the incidence rate increased by 9.3% during the period (Table 1, Appendix A).

We observed a great decrease in the prevalence (Figure 1a) and incidence (Figure 1b) of HBV, starting in 2000, with significant changes in 2000 and 2004. For HCV, there were five significant changes. Conversely, although the prevalence and incidence of HCV had a significant decrease until 2004, from this year began a period of increase, with an APC of 1.46% from 2014 to 2019 for prevalence, and 0.81% for incidence (Figure 1).

The global AAPC for HBV (−1.3%) was 3.25 times that for HCV (−0.4%) during the study period. These ratios differed between SDI categories. The highest ratio was in low–middle-SDI countries (4.00), and the lowest was in middle-SDI countries (1.89). The highest AAPC for HBV occurred in middle-SDI countries, and the lowest was in high-SDI countries; this was similar for HCV (Appendix A, Figure 2).

### 3.2. HBV and HCV Mortality

In 2019, 1.09 million deaths occurred from HBV or HCV (95% UI 0.96–1.23), including all ages and both sexes worldwide; 555.4 thousand [95% UI 487.1–630.1] for HBV and 542.3 thousand [95% UI 476.7–608.8] for HCV. The ASMR per 100,000 persons was 7.18 (95% UI 6.30–8.14) and 7.01 (95% UI 6.16–7.87) for HBV and HCV, respectively.

The HBV and HCB total percentage changes for the ASMR (2019/1990) were −26.7% and 10.0%, respectively. While ASMR for HBV decreased, it increased for HCV during this period. The percentage change in ASMR for HBV was highest in countries with high–middle SDI and lowest in countries with high SDI. For HCV, the percentage change in ASMR was highest in countries with high SDI (increase), and it decreased only in countries with low SDI (Table 2).

Appendix A shows the APC and AAPC of ASMR in HBV and HCV for all ages and both sexes from 1990 to 2019, for global trends as well as for countries grouped by SDI.

We observed a great decrease in HBV mortality in high–middle- and middle-SDI countries, especially between 2000 and 2004 (APC = −6.15) (Figure 3a). In contrast, HCV mortality (Figure 3b) increased for high-SDI countries, especially between 1990 and 1999 (APC = 2.51–2.95) and decreased significantly for low-SDI countries between 1999 and 2015 (APC = −1.03, −1.43) (Figure 3).

## 4. Discussion

Our analysis showed some encouraging results because incidence and prevalence of HBV and HCV have been decreasing around the world between 1990 and 2019. In contrast, mortality has decreased for HBV but not for HCV, which is somewhat disappointing. Despite the advances, there is still a long way to go to achieve the 2030 goal: a reduction of 90% in incidence and 65% in mortality for both infections compared to the 2015 baseline. An important challenge is related to finding a way to speed up the yearly rate at which the decline is happening.

Our data and analysis have some limitations. They are based on data from the Global Burden of Disease repository, which is curated by the Institute of Health Metrics at the University of Washington and has made huge efforts to improve the quality of the information. However, there are still major challenges for information quality, including differences in the health coverage services between countries, quality and coverage of death certificates, and accuracy of modeling strategies dealing with inconsistencies in information between countries. Those aspects may influence the differences observed between developed and developing countries because part of the differences may be due to data quality rather than to diseases behaving in different ways [15]. For instance, non-alcoholic fatty liver diseases (NAFLD) have been rising fast as an emergent cause of chronic liver disease, especially in developing countries, and it is likely that, for countries with weaker health care systems, a proportion of cases of NAFLD may be misclassified as chronic liver diseases due to viral hepatitis B or C [16].

Most of the achievements in the reduction in hepatitis B incidence are due to the use of a highly effective vaccine early in childhood. In fact, some WHO regions, like Europe (EURO) and the Americas and the Caribbean (PAHO), have achieved the goals of eliminating mother-to-child transmission [17]. However, there are fewer advances for the treatment and cure of the people who were infected before the vaccine was available, which amounts to an estimate of 257 million people. Current available treatments for hepatitis B virus may stop virus replication, but they cannot eliminate the virus from a carrier [18,19].

The largest numbers of chronic hepatitis B carriers are located in WHO’s Western Pacific Region (116 million), African Region (82 million), and South Pacific Region (60 million) [20]. Therefore, advances in elimination in those regions are critical for the fulfillment of the global elimination plan. China is probably the single country with the most carriers—with approximately 70 million people carrying HBsAg—and it has made significant advances in mother-to-child infection prevention. Vaccination coverage among children born from HBeAg+ mothers is above 95%, but challenges still remain ahead [21]. The most challenging aspect of this problem is how to conduct the massive screening required to identify HBsAg+ carriers for treatment; like other countries in the Western Pacific Region, China has committed to identifying 30% of the people living with HBV and treating 50% of them [22]. A recent study has found that, between 2010 and 2018, the rate of detection of HBV positive increased from 5% to 19%, and the rate of antiviral therapy among those eligible increased from 4% to 31%. However, that level of increase is not enough to reach the described goals for 2030 [23].

Our analysis showed that, in 2019, mortality for hepatitis B had reduced by 28% compared to the 2010 baseline. Most of the reductions have been linked to prevention measures like vaccination, ensuring safer blood transfusion, preventing sexual transmission, etc. However, there are still more than 500,000 deaths every year due to HBV, and most of them occur in China, which reports more than 400,000 cases of hepatocellular carcinoma (HCC) per year [24]. So, it is imperative to try to improve the survival of HCC patients to further reduce the number of HBV deaths, but there are many barriers to doing this [25].

For hepatitis C control, there is a lack of a prophylactic measure, and advances in prevention have been linked to improved screening efforts at blood banks, which have greatly reduced the main source of infections for many populations. However, this measure alone is unlikely to eliminate hepatitis C transmission since there is a residual risk of infection of 1 in 100,000 transfusions [26]. Recently, there have been advances in the discovery of new antivirals that can eliminate the infection from the liver and have the potential to save lives and boost the decrease in incidence. However, high prices have deterred many low- and middle-income countries from introducing massive searches and treatment programs for hepatitis C. WHO has estimated that it would need an investment of about USD 57 billion to eliminate viral hepatitis in 67 countries [27].

The European Union HCV Collaborators conducted a study assessing the prevalence of HCV in the EURO Region and the level of intervention required to achieve WHO targets for HCV elimination. Its final conclusion was that, despite its advanced healthcare infrastructure, the expansion of screening programs and diagnosis should increase from 88,800 new cases annually in 2015 to 180,000 in 2025 to boost treatment in order to achieve the WHO’s targets [28]. However, Wedemeyer H. et al. believe that, as of 2021, most countries are not on track to reach the 2030 HCV elimination targets set by the WHO. Moreover, the COVID-19 pandemic resulted in a decrease in HCV diagnoses and fewer direct-acting antiviral treatment initiations in 2020 [29].

In contrast, Rockstroh JK. et al. consider that, except in a few countries (e.g., Egypt), elimination targets are endangered in the majority of the world—regardless of income level [30]—and they state that this is largely because the greater burden of HCV infection is carried by marginalized populations of people who use drugs (PWUD) and people who are incarcerated, who are being left behind in HCV control efforts [31]. According to data from the Polaris observatory (https://cdafound.org/polaris-regions-database/ (accessed on 17 August 2023)), Australia has detected more than 90% of HCV-infected people and is on track to eliminate that virus. Most other continents and countries have detected less than 60%, and low- and middle-income countries less than 30%.

The small yearly pace at which both hepatitis agents are reducing in incidence and mortality signals that it is unlikely that elimination goals can be met by 2030. For hepatitis B, the use of vaccines must be supplemented with other measures such as screening and treating pregnant HBsAg+ mothers to reduce the number of children infected at birth and identifying chronic-carrier adults and treating them to reduce mortality. However, as in the case of treating hepatitis C, most countries cannot bear the costs of doing those actions or do not have in place the technical capacity to do so. In contrast, hepatitis B vaccine effectiveness is reduced because there are many countries that have not introduced the birth dose [32,33].

Although the elimination goals were reformulated [34], the following problems and barriers to their achievement persist, especially in low-income countries: (a) low technical capability to identify the population carrying the virus; (b) low access to treatment with direct-acting antivirals in people diagnosed; and (c) high financial impact on health systems’ budgets. Thus, hepatitis B and C elimination may be achieved only if novel mechanisms of financing the cost of antivirals are devised or if a dramatic decrease in price can be negotiated with the industry.

## 5. Conclusions

The global HBV and HCV rates have fallen due to differences in AAPC linked to the SDI. Despite the advances made, there is still a long way to go to achieve the 2030 elimination goals. An important challenge is related to finding a way to speed up the yearly rate at which the decline is happening, which may include reducing the costs of hepatitis C antivirals and finding better ways to implement population screening to find and treat hepatitis B or C carriers.

## Figures and Tables

**Figure 1 microorganisms-12-00388-f001:**
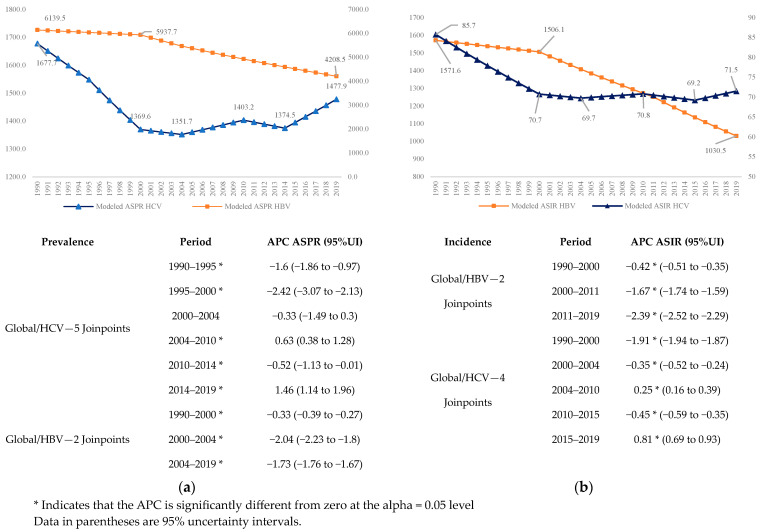
Global trends in HBV and HCV infection prevalences and incidences for all ages and both sexes 1990–2019: (**a**) Age-Standardized Prevalence Rate 1990–2019; (**b**) Age-Standardized Incidence Rate 1990–2019.

**Figure 2 microorganisms-12-00388-f002:**
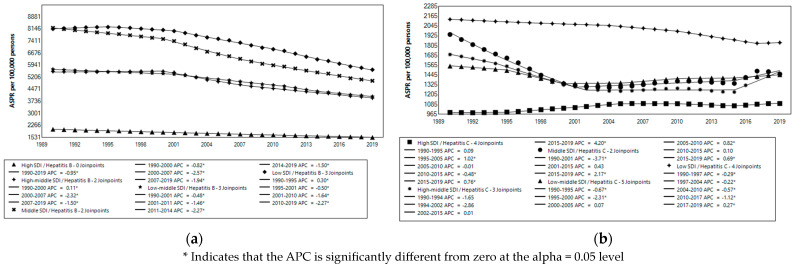
Trends in HBV and HCV infection prevalences 1990–2019: (**a**) Trends in HBV Prevalence 1990–2019 in countries grouped by SDI; (**b**) Trends in HCV Prevalence 1990–2019 in countries grouped by SDI.

**Figure 3 microorganisms-12-00388-f003:**
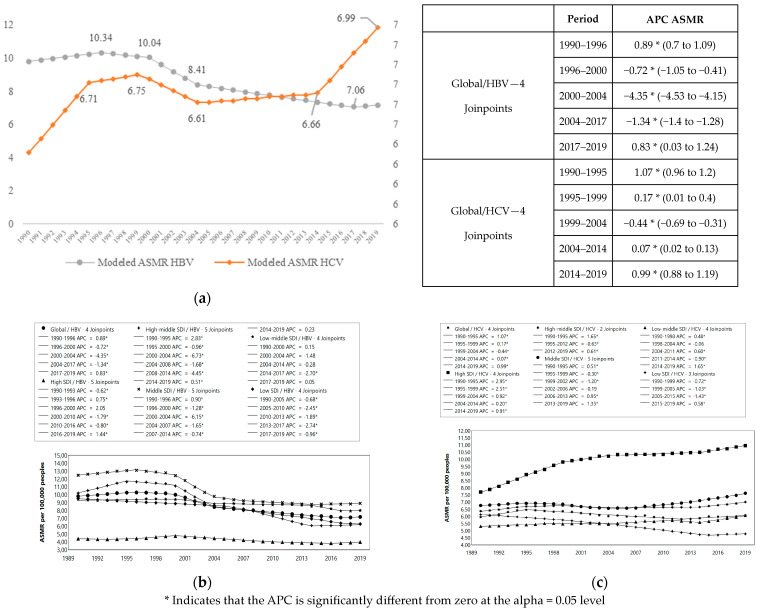
Trends in HBV and HCV Mortality 1990–2019: (**a**) Trends in HBV and HCV mortality 1990–2019 worldwide; (**b**) Trends in HBV mortality 1990–2019 in countries grouped by SDI; (**c**) Trends in HCV mortality 1990–2019 in countries grouped by SDI.

**Table 1 microorganisms-12-00388-t001:** Percentage change in age-standardized Hepatitis B and C prevalence and incidence rates by SDI category, for all ages and both sexes, 1990–2019.

Countries	ASR—Hepatitis B	Percentage Change ASR-HBV (2019/1990)	ASR—Hepatitis C	Percentage Change ASR-HCV (2019/1990)	Ratio Percentage Change ASR Hep B/C
**Prevalence**					
**Global**					
1990	6145.4 (5489–6787.3)		1677.3 (1362.8–2045.4)		
2019	4216.3 (3776.3–4667.2)	−31.4%	1463.3 (1189–1796.6)	−12.8%	2.5
**High SDI**					
1990	2027.1 (1860.4–2190.4)		986.2 (820.9–1181.5)		
2019	1531.6 (1397.1–1649)	−24.4%	1096.2 (914.4–1318.1)	11.2%	−2.2
**High–middle SDI**					
1990	5546.3 (4994.2–6119.5)	−29.1%	1695.1 (1377.2–2060)	−14.0%	2.1
2019	3935 (3520–4364.1)		1457.1 (1186.3–1783.4)		
**Middle SDI**					
1990	8209.2 (7353.1–9076.2)		1940.1 (1576.1–2378.1)		
2019	4982.4 (4470.2–5502.9)	−39.3%	1455.7 (1174.7–1800)	−25.0%	1.6
**Low–middle SDI**					
1990	5685.4 (5004.9–6375.4)		1553.5 (1260.4–1903.7)		
2019	4027.3 (3594.9–4465.9)	−29.2%	1447.6 (1176.8–1784)	−6.8%	4.3
**Low SDI**					
1990	8146.1 (7125.1–9170.2)		2125.2 (1719.1–2636.3)		
2019	5656.9 (4995.8–6338.6)	−30.6%	1840.7 (1491.1–2293.4)	−13.4%	2.3
**Incidence**					
**Global**					
1990	1569.14 (1288.28–1878.36)		85.61 (75.54–98.65)		
2019	1034.23 (827.14–1270.32)	−34.1%	71.27 (62.8–82.55)	−16.8%	2.04
**High SDI**					
1990	620.76 (517.9–729.78)		49.85 (43.78–57.7)		
2019	408.18 (332.29–488.4)	−34.2%	54.5 (47.02–63.05)	9.3%	3.67
**High–middle SDI**					
1990	1364.38 (1104.74–1632.81)		68.5 (60.48–78.98)		
2019	823.64 (612.07–1048.06)	−39.6%	50.11 (43.56–58.67)	−26.8%	1.48
**Middle SDI**					
1990	2047.46 (1676.62–2458.51)		92.09 (81.04–106.21)		
2019	1186.58 (928.12–1462.98)	−42.0%	67.29 (59.38–77.91)	−26.9%	1.56
**Low–middle SDI**					
1990	1524.96 (1251.58–1846)		91.29 (80.1–105.86)		
2019	1089.32 (873.27–1332.33)	−28.6%	72.43 (63.57–84.84)	−20.7%	1.38
**Low SDI**					
1990	2033.1 (1680.18–2442.64)		145.29 (125.61–171.21)		
2019	1454.42 (1187.71–1769.41)	−28.5%	119.76 (103.52–142.23)	−17.6%	1.62

Data in parentheses are 95% uncertainty intervals.

**Table 2 microorganisms-12-00388-t002:** Percentage change in ASMR of HBV and HCV by SDI-grouped countries, all ages and both sexes, 1990 and 2019.

SDI/YEAR	ASMR-HBV	Percentage Change ASMR-HBV (2019/1990)	AAPC ASMR-HBV 1990–2019	ASMR-HCV	Percentage Change ASMR-HCV (2019/1990)	AAPC ASMR-HCV 1990–2019
Global						
1990	9.8 (8.75–10.94)	−26.7%	−1.1 * (−1.1, −1.0)	6.37 (5.62–7.13)	10.0%	0.3 * (0.3, 0.3)
2019	7.18 (6.3–8.14)			7.01 (6.16–7.87)		
High SDI						
1990	4.42 (3.91–5.00)	−9.5%	−0.3 * (−0.4, −0.3)	7.71 (6.98–8.52)	42.2%	1.2 * (1.2, 1.3)
2019	4.00 (3.48–4.59)			10.96 (9.75–12.11)		
High–Middle SDI						
1990	10.19 (8.94–11.61)	−38.7%	−1.7 * (−1.8, −1.7)	5.96 (5.25–6.7)	2.0%	0.1 * (0.0, 0.1)
2019	6.25 (5.36–7.13)			6.08 (5.34–6.86)		
Middle SDI						
1990	12.47 (11.1–13.98)	−28.6%	−1.2 * (−1.2, −1.1)	6.77 (5.98–7.59)	12.7%	0.4 * (0.4, 0.4)
2019	8.9 (7.72–10.12)			7.63 (6.52–8.77)		
Low–Middle SDI						
1990	9.41 (8.16–10.82)	−15.0%	−0.5 * (−0.6, −0.5)	5.32 (4.49–6.22)	14.1%	0.5 * (0.4,0.5)
2019	8.00 (6.85–9.24)			6.07 (5.15–7.08)		
Low SDI						
1990	9.52 (7.8–11.67)	−34.0%	−1.4 * (−1.4, −1.4)	6.13 (5.08–7.42)	−22.0%	−0.9 * (−0.9, −0.8)
2019	6.28 (5.23–7.47)			4.78 (4.01–5.69)		

Data in parentheses are 95% uncertainty intervals. * Indicates that the AAPC is significantly different from zero at the alpha = 0.05 level.

## Data Availability

Data were taken from the GBD 2019, a publicly available online resource “https://vizhub.healthdata.org/gbd-results/” (accessed on 17 August 2023) by the Institute for Health Metrics and Evaluation (IHME) at the University of Washington [11].

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
