# Peer review of "How Possible Is the Elimination of Viral Hepatitis? An Analysis Based on the Global Burden of Disease from Hepatitis B and C, 1990–2019"

_microorganisms, 2024, doi:10.3390/microorganisms12020388_

Round 1

Reviewer 1 Report

Comments and Suggestions for Authors

Important topic, nice presentation, significant conclusions. The burden of viral hepatitis seems to decrease, however, it seems to be proven that the original aim of WHO will not be fulfilled by 2030.

 The text of the manuscript is entirely OK. There are only several minor errors to be corrected. The Sociodemographic Index (SDI) categories in the Materials and Method section were mentioned as High, Medium-High, Medium, Medium-Low and Low. However, in the Abstract and Tables 1 and 2 and Table S1 and S2 these categories mentioned as High-middle, Middle, Middle, Low-middle (or middel). These must be unified in the final version. 

I don't mind which version, but should be the same. Comments on the Quality of English Language

Some minor changes are needed, please correct middle-middel etc.

Author Response

Dear Reviewers

Here are our responses to your comments

Reviewer 1

Important topic, nice presentation, significant conclusions. The burden of viral hepatitis seems to decrease, however, it seems to be proven that the original aim of WHO will not be fulfilled by 2030.

The text of the manuscript is entirely OK. There are only several minor errors to be corrected. The Sociodemographic Index (SDI) categories in the Materials and Method section were mentioned as High, Medium-High, Medium, Medium-Low and Low. However, in the Abstract and Tables 1 and 2 and Table S1 and S2 these categories mentioned as High-middle, Middle, Middle, Low-middle (or middle). These must be unified in the final version. 

I don't mind which version but should be the same.

R/. We welcome your comments and annotations. And the classification of countries according to SDI has been corrected, as it appears in reference 11.

Reviewer 2 Report

Comments and Suggestions for Authors

In this study, authors analyzed the data collected from a public database to investigate trends in prevalence, incidence and mortality from HBV and HCV infections worldwide for 1990-2019. ASIR, ASPR and ASMR changes for HBV and HCV were calculated. The results were presented comprehensively. There are some issues that should be addressed.

1. The title “How possible is the elimination of viral hepatitis?” is too vague. Since authors only analyzed the prevalence data of HBV and HCV, it is suggested that the title should be adjusted.

2. Line 77: It is recommended that more information of SDI should be provided.

3. Line 82: Authors claimed that they collected data from GBD 2019. Can authors provide more details of this database? How many countries’ data were included?

4. In the Discussion section, it is suggested that authors can explain more about the differences between prevalence trends in HBV and HCV. The current version mainly focuses on HCV.

Comments on the Quality of English Language

Minor editing of English language required

Author Response

Dear Reviewers

Here are our responses to your comments

Reviewer 2

Comments and Suggestions for Authors

In this study, authors analyzed the data collected from a public database to investigate trends in prevalence, incidence and mortality from HBV and HCV infections worldwide for 1990-2019. ASIR, ASPR and ASMR changes for HBV and HCV were calculated. The results were presented comprehensively. There are some issues that should be addressed.

  1. The title “How possible is the elimination of viral hepatitis?” is too vague. Since authors only analyzed the prevalence data of HBV and HCV, it is suggested that the title should be adjusted.

R/. We welcome your comments and annotations. As far as the title is concerned, we believe that it is appropriate given that we have not only examined the prevalence but also the incidence and mortality for both hepatitis C and hepatitis B. However, another possible title could be: "is it possible to eliminate viral hepatitis B and C by 2030?"

  1. Line 77: It is recommended that more information of SDI should be provided.

R/. The significance of the SDI as an indicator is expanded:  The Sociodemographic Index (SDI) regions (High, Middle-high, Middle, Middle-low and Low) measuring 195 countries and territories grouped according to the geometric mean of three components: a) Total fertility rate under the age of 25 (TFU25); b) Mean education for those ages 15 and older (EDU15+); and c) Lag distributed income (LDI) per capita [1]. The SDI is a composite indicator of development status that is strongly correlated with health outcomes. It is used in the Global Burden of Disease (GBD) study to help produce estimates of the burden of diseases, injuries, and risk factors for various locations [1].

  1. Line 82: Authors claimed that they collected data from GBD 2019. Can authors provide more details of this database? How many countries’ data were included?

R/. The GBD 2019 estimates incidence, prevalence, mortality, years of life lost (YLLs), years lived with disability (YLDs), and disability-adjusted life-years (DALYs) due to 369 diseases and injuries, for two sexes, and for 204 countries and territories. Input data were extracted from censuses, household surveys, civil registration and vital statistics, disease registries, health service use, air pollution monitors, satellite imaging, disease notifications, and other sources [1].

  1. In the Discussion section, it is suggested that authors can explain more about the differences between prevalence trends in HBV and HCV. The current version mainly focuses on HCV.

R/. We welcome your comments. New paragraph in discussion:

The largest amount of chronic hepatitis B carriers are located at WHO’s Western Pacific Region (116 million), African Region (82 million), and South Pacific Region (60 million) [20]. Therefore, advances in elimination in those regions are critical for the fulfillment of the global elimination plan. China is probably the single country with more carriers -approximately 70 million people carrying HBsAg- and has made significant advances in mother to child infection prevention. Vaccination coverage among children born from HBeAg+ mothers is above 95%, but challenges still remained ahead. [21] The most challenging aspect is how to conduct massive screening to identify HBsAg+ carriers and treat them; and, as other countries in the Western Pacific Region, China has committed to identify 30% of the people living with HBV and treating 50% of them. [22]. A recent study has found that, between 2010 and 2018, the rate of detec-tion of HBV positive have increased from 5% to 19%, and the rate of antiviral therapy among those eligible increased from 4% to 31%. However that level of increase is not enough to reach the goals for 2030 [23]

References

  1. Vos T, Lim SS, Abbafati C, Abbas KM, Abbasi M, Abbasifard M, et al. Global burden of 369 diseases and injuries in 204 countries and territories, 1990–2019: a systematic analysis for the Global Burden of Disease Study 2019. The Lancet. 2020 Oct 17;396(10258):1204–22.

Reviewer 3 Report

Comments and Suggestions for Authors

The authors discuss the important topic of possible elimination of HBV and HCV infections. The topic is important and interesting. The manuscript is well written and the limitations are correctly identified and described in the discussion. I propose that the manuscript be accepted for publication

Author Response

Dear Reviewers

Here are our responses to your comments

Reviewer 3

The authors discuss the important topic of possible elimination of HBV and HCV infections. The topic is important and interesting. The manuscript is well written and the limitations are correctly identified and described in the discussion. I propose that the manuscript be accepted for publication

R/. Thank you very much by your comments.

Round 2

Reviewer 2 Report

Comments and Suggestions for Authors

The authors have revised the manuscript accordingly.